# Prevalence and Awareness of Hypertension among a Rural Jazan Population

**DOI:** 10.3390/healthcare11121676

**Published:** 2023-06-07

**Authors:** Luai Alhazmi, Maged El-Setouhy, Alhassan H. Hobani, Raed E. Jarram, Mohsen J. Zaylaee, Rakan S. Hazazi, Mohammed A. Nasib, Ammar A. Musawa, Atheer Y. Hakami, Mohamed S. Mahfouz, Omar Oraibi

**Affiliations:** 1Department of Medicine, Faculty of Medicine, Jazan University, Jazan 45142, Saudi Arabia; 2Department of Family and Community Medicine, Faculty of Medicine, Jazan University, Jazan 45142, Saudi Arabia; 3Faculty of Medicine, Jazan University, Jazan 45142, Saudi Arabia201808877@stu.jazanu.edu.sa (M.J.Z.);

**Keywords:** hypertension, awareness, rural areas, Jazan, Saudi Arabia

## Abstract

Background: Hypertension (HTN) is a major global public health problem. Knowledge of the risk factors and repercussions of HTN is crucial to preventing the disease. Rural populations have lower levels of knowledge of the disease than urban populations. However, no studies have assessed the levels of awareness of HTN and their determinants in rural regions of Saudi Arabia. Objectives: This study aimed to assess the awareness of HTN and its determinants among a rural population of Jazan region, Saudi Arabia. Methodology: We conducted a cross-sectional analytical study among six primary healthcare centers selected randomly from the rural areas of Jazan region. We targeted all Saudi adults visiting these centers. Information was gathered using interview questionnaires completed by 607 people. SPSS was utilized to analyze the collected data. Results: In all population groups, the prevalence of diagnosed HTN increased with age, particularly gradually increasing in those aged younger than 40 years and then rapidly and sharply increasing in those aged 40 years and over. The women (43.3%) had a higher prevalence of HTN than the men (34.6%), which is comparable with findings in other areas in Saudi Arabia and the Middle East. Approximately 65.6% of the participants without HTN and 34.4% of the participants with HTN did not know their normal blood pressure. Approximately 61.7% of the participants without HTN and 59.0% of the participants with HTN felt that pharmaceuticals are insufficient in curing HTN, while 60.7% and 64.7% believed that HTN can be cured. Conclusions: The global prevalence of HTN is increasing annually owing to rapid changes in lifestyle and dietary habits. Furthermore, because adherence to antihypertensives is poor in rural Jazan, the Ministry of Health and researchers advocate implementing a program to increase awareness and assess patient adherence to prescribed medication for the control of HTN.

## 1. Introduction

Hypertension (HTN) in adults is a substantial global public health issue that contributes to a high disease burden. It occurs when the systolic blood pressure (BP) is at least 140 mmHg and/or the diastolic BP is at least 90 mmHg [1,2]. In a smaller proportion of cases, HTN has a definite cause, whereas in the majority of cases, the cause is unknown, yielding the use of the term “essential HTN” [3]. Essential HTN is a mysterious disease associated with a variety of risk factors, including age, overweight, family history, and ethnicity. It is also affected by dietary and lifestyle habits. Salt sensitivity has long been believed to increase the likelihood of developing HTN; approximately 50–60% of individuals are sensitive to salt and therefore develop HTN [4,5]. With accelerated changes in lifestyle and dietary patterns, the prevalence of HTN worldwide is rising annually, with more than 30% of the population affected [1,6]. Uncontrolled HTN can lead to a number of life-threatening complications, such as heart, kidney, and brain diseases, which mostly result in disability [7].

As the global prevalence of HTN continues to rise, increasing awareness and knowledge of HTN has become a top priority in the fight against cardiovascular diseases [8]. In recent years, substantial progress has been made in raising awareness of and detecting HTN; however, some individuals still have low levels of knowledge and awareness about HTN. Even some patients with HTN lack adequate knowledge regarding BP control [9]. The prevalence of HTN is 26.1% among Saudis [10]. The awareness about the disease is also increasing [11]. However, in Jazan region, which is mostly rural than urban, the prevalence of the disease ranges between 7.5% and 9% according to the Household Health Survey [12]. Studies indicate that the level of awareness of HTN is typically lower in rural areas than in urban areas [13,14]. Therefore, we hypothesized that there is a knowledge deficit regarding HTN among residents of Jazan region. Accordingly, this study was conducted in Jazan to determine the level of awareness regarding HTN among an adult rural population and the need to establish programs that increase community awareness of the long-term complications of uncontrolled HTN and proper BP control practices to improve self-care practices and BP monitoring among patients with HTN in rural areas.

## 2. Methodology

### 2.1. Study Setting

The Jazan region is one of the 13 regions of Saudi Arabia, which lies in the far southwest of the Kingdom adjacent to Yemen. We conducted this study in Jazan, as this region is mostly rural than urban. The region has 177 rural primary healthcare centers (PHCCs) serving over a million inhabitants [15].

### 2.2. Study Design and Subject Selection

We utilized a cross-sectional analytical study design. The study was conducted in six PHCCs selected randomly from the rural Jazan region. We targeted all male and female adult Saudis who visited these centers between December 2021 and January 2022. Adult Saudis with cognitive impairments, those who were unable to respond to the questionnaire, and those who used medications that might have influenced the results were excluded.

### 2.3. Sample Size and Data Collection

The sample size was estimated using the following statistical equation: initial sample size = [(z^2^ × p × q)]/d^2^. After informed consent was obtained, we designed an interview questionnaire, piloted it, and used it to collect data. A total of 27 questions were included in the survey, which was divided into two sections. The first section contained items on sociodemographic information, such as age, sex, educational level, income, and marital status. The second section was designed to examine the knowledge and awareness of HTN of the participants. After permission was obtained from the participants, the PHCC nurse gathered data on BP by mercury sphygmomanometer, height, and weight. A sample of 607 Saudis visiting certain PHCCs was determined using a confidence interval (CI) of 95% and a margin of error of less than 5%. The random sampling technique was employed in three stages. In the first stage, we randomly selected six PHCCs from a total of 177 centers in rural Jazan. In the second stage, we randomly selected one day per week to collect our data. Finally, all Saudi adults visiting the selected PHCCs during the selected days were invited to participate in the study.

### 2.4. Ethical Considerations

The study was presented primarily to the Scientific Research Ethics Committee of the Ministry of Health for review and approval. The study was approved with reference number 2194. It was conducted following ethical principles in Saudi Arabia, and informed consent was obtained from each participant before starting the anonymous questionnaire survey. The participants were free to leave the survey anytime during the research process. None of the participants was questioned about anything that could disclose their identity; privacy and confidentiality were maintained.

### 2.5. Statistical Analysis

Data were coded and entered into a Microsoft Excel sheet (version 2016). The IBM SPSS Statistics for Windows (version 25.0, Armonk, NY, USA) was used for the statistical analysis. Descriptive and inferential statistics were used for the data analysis. Frequencies and percentages were used to summarize the categorical data. The chi-squared test was performed to determine the associations between individual categorical variables and hypertritons. Significant variables from the chi-squared test were included in the logistic regression analysis. Logistic regression models were used to identify the factors affecting the prevalence of HTN. Both bivariate and multivariate logistic regression models were utilized. Crude odds ratios and adjusted odds ratios (aORs) with their 95% CIs were estimated. Finally, *p*-values of <0.05 indicated statistical significance.

## 3. Results

### 3.1. Sociodemographic Characteristics

The study included a total of 607 individuals, among whom 59.6% (n = 362) were men, and 40.4% (n = 245) were women. Out of the 607 participants, 210 were hypertensive, resulting in a prevalence of 12% (28.7% for the men and 43.3% for the women). The prevalence of HTN increased with age. The participants aged 65 years and above had the highest prevalence of HTN (82.7%). HTN was more common among the illiterates (70.1%) than among their counterparts. A total of 363 participants were earning more than SAR 5000 per month, whereas only 49 were earning more than SAR 15,000 per month. The widowed participants (82.9%) had a higher prevalence of HTN than the married participants (36.1%). Approximately 79.9% of the participants were nonsmokers; only 15.5% were current smokers; and 4.6% were ex-smokers (Table 1).

### 3.2. Knowledge of the Normal BP and Awareness of HTN

The awareness about HTN was assessed using validated questionnaires in 397 participants without HTN and 210 participants with HTN (Table 2). The normal BP was unknown to 56.6% of the participants without HTN and 56.1% of the participants with HTN. Approximately 61.7% of the participants without HTN and 59.0% of the participants with HTN believed that medications are insufficient in curing HTN, whereas 60.7% and 64.7% believed that HTN is curable. Regarding lifestyle changes, 21.9% of the participants without HTN and 29.0% of the participants with HTN believed that changing one’s lifestyle has no influence on BP control. HTN was considered harmful and to progress to other life-threatening diseases by 80.1% of the participants without HTN and 84.2% of the participants with HTN. The participants with and without HTN had different levels of knowledge and awareness of the HTN risk factors. Smoking was considered a risk factor for HTN by 476 participants, obesity by 531 participants (86.9% of the participants without HTN and 88.5% of the participants with HTN), and stress by 449 participants (73.8% of the participants without HTN and 74.2% of the participants with HTN).

### 3.3. Determinants of HTN

Table 3 shows the determinants of HTN among the study groups classified according to sex, age, educational level, income, and marital status. According to sex, 321 male participants (88.7%) considered obesity as a risk factor for HTN, while 225 female participants (91.8%) considered salt consumption as a risk factor for HTN. All age groups believed salt consumption to be a risk factor for HTN: 18–24 years (n = 43, 69.4%), 25–44 years (n = 256, 90.1%), 45–64 years (n = 164, 91.1%), and 65 years and above (n = 75, 92.6%). In addition, the 18–24-year age group (n = 43, 69.4%) considered obesity as a risk factor for HTN. According to the educational level, the illiterates (n = 99, 92.5%), participants with secondary school education (n = 150, 86.7%), and participants with high school education (n = 185, 92.5%) considered salt consumption as a risk factor for HTN. In contrast, the participants with primary school education (n = 107, 84.2%) identified obesity as a risk factor for HTN. In terms of income, the participants earning SAR <5000 (n = 316, 87.1%), 5000–9999 (n = 110, 88%), and ≥15,000 per month (n = 48, 98%) believed salt consumption to be a risk factor for HTN. In contrast, the majority of the participants earning SAR 1000–14,999 per month identified obesity as a risk factor for HTN. All groups according to the marital status considered salt consumption as a risk factor for HTN: single (n = 118, 81.4%), married (n = 367, 90.8%), divorced (n = 21, 91.3%), and widowed (n = 32, 91.4%).

We investigated the factors associated with HTN using univariate and multivariate logistic regression analyses. The univariate analysis revealed that age, sex, educational level, income, marital status, and tobacco chewing were significantly associated with HTN. The multivariate logistic regression analysis suggested that the most important independent predictors among our sample were age, sex, and income. The likelihood of HTN increased with advancing age, with a 13% increase for each year (aOR = 1.13, 95% CI = 1.10–1.16, *p* < 0.001). The female participants were more likely to be hypertensive than the male participants (aOR = 1.79, 95% CI = 1.06–3.03, *p* = 0.029). The participants earning SAR 5000–9999 and those earning SAR 15,000 and above per month were 51% (aOR = 0.49, 95% CI = 0.26–0.93, *p* = 0.029) and 64% (aOR = 0.36, 95% CI = 0.14–0.92, *p* = 0.023) less likely to have HTN, respectively, than those earning less than SAR 5000 per month (Table 4).

## 4. Discussion

HTN is considered as one of the main predictors of cardiovascular morbidity and mortality in numerous countries worldwide [16]. According to previous studies, the prevalence of HTN in Saudi Arabia is rising in line with the global trend. However, limited research has been conducted on the level of HTN awareness in Middle Eastern nations in general and Saudi Arabia in particular [1,17]. In addition, the majority of these studies have focused on urban areas to reduce the prevalence of and manage HTN. There is no specific information about the prevalence or level of awareness of HTN in the rural Jazan region, so we decided to conduct this study to achieve a better understanding and provide evidence for health decision-makers to improve and ensure that lacking areas are well covered [18]. In this study, we discovered that the women (43.3%) had a higher prevalence of HTN than the men (28.7%) in the rural Jazan region, which is consistent with findings in other Saudi Arabian and international studies [1,19]. A similar study undertaken in the Middle East reported that women had a higher prevalence of HTN than men, with 24.5% of men and 24.7% of women having HTN in Egypt, 31.2% of men and 33.6% of women in Morocco, and 28% of men and 37.7% of women in Qatar [20]. Herein, we also discovered that the prevalence of HTN was inversely related to wealth, which contradicts previous reports [19,21]. This finding might be attributed to lifestyle habits as well as the lack of resources and financial constraints for HTN treatment.

Our research found that both participants with (56.1%) and without HTN (56.6%) in rural areas were undereducated and did not know the normal BP. In a similar study conducted in Sri Lanka, 53.6% of patients with HTN were uninformed of the normal BP. These findings show that knowledge of the normal BP is lacking and thus has to be greatly improved [22]. These findings imply that the general public, particularly patients with HTN, should be educated about the necessity of understanding the appropriate BP. The level of knowledge about medications being insufficient in controlling high BP was high in our study, with 61.7% of the participants without HTN and 59.0% of the participants with HTN agreeing. These proportions are lower than the 77.5% of patients with HTN in Saudi Arabia who believe that their BP can decrease with appropriate treatment according to a recent survey [23]. In the present study, 60.7% of the participants without HTN and 64.7% of the participants with HTN agreed that HTN is a treatable disease. A similar finding was reported in a research conducted in Sri Lanka, with 68.6% believing that HTN can be treated [22].

In our study, 80.1% of the participants without HTN and 84.2% of the participants with HTN were aware of the complications of HTN and that HTN can lead to other life-threatening diseases. According to a previous study, only 23.75% of people were aware of the complications associated with HTN [9]. The rural Jazan adults in our study were more aware about the HTN risk factors than patients in other studies conducted worldwide [24,25,26]. The most common risk factors reported by our participants were increased salt consumption (90.9%), obesity (88.5%), smoking (75.2%), and stress (74.2%). According to prior research conducted in Al-Ahsa, Saudi Arabia, 90% of participants felt that excessive salt consumption was a risk factor; 89.1%, stress; 75.5%, obesity; and 62.1%, smoking [27]. Another study performed in Riyadh, Saudi Arabia, found that 84.4% and 87.5% of participants agreed that excessive salt consumption and obesity were risk factors for HTN, respectively [28]. Finally, the levels of awareness of HTN were worse in rural areas than in urban areas according to several studies conducted in Saudi Arabia and around the world [10,13,14,29].

Many studies have been conducted to understand the knowledge of the population about the normal BP and whether there is any significant difference between such knowledge and sex, age, educational level, or income. The results implied that individuals in the low-income class were less likely to be aware of the normal BP than their counterparts. In contrast, individuals who were in the high-income class, were middle-aged, and had higher educational levels had a better level of knowledge of the normal BP. The findings are consistent with previous reports in Al-Ahsa [27], China [30], Nigeria [31], and rural Nepal [32].

In many communities, marriage is an important upstream determinant that influences the awareness of HTN. Therefore, the level of awareness of HTN remained low among individuals with infrequent contact with a partner and single or divorced adults. A previous study conducted in China demonstrated that married patients with HTN were more likely to be aware of their BP condition than single patients [30]. Manfredini et al. observed that married individuals had good control of their BP and were aware of their HTN, unlike single or divorced patients [33]. The most possible explanation for this difference is that married patients with HTN are more likely to receive support from their partners and health professionals about their health issues than unmarried patients [33,34,35,36]. Interestingly, in our study, we found that there was no significant difference between the married and single participants.

Herein, the middle-aged population (25–44 years) had a higher level of knowledge about most risk factors associated with HTN than the other populations. In contrast, the young participants (18–24 years) had the lowest level of knowledge about the risk factors of HTN. This finding is consistent with a prior report in China among patients with HTN [30]. This consistency might be attributed to the lower attention paid to health as well as the lower probability of having HTN at this age.

In our study, we found a significant association between the level of education and the level of knowledge about HTN. The majority of the participants with high school education had the highest level of knowledge about the risk factors of HTN. Such findings were also reported in prior studies, suggesting that individuals with high levels of education and high family incomes have easier and better contact with healthcare providers as well as access to information technologies (e.g., television, computer), both of which predictably contribute to higher levels of awareness [11,27,37,38].

### Study Limitations

Although this study is one of the few studies to evaluate the awareness of patients with HTN in Saudi Arabia in general and the Jazan region in particular, it has some limitations that should be addressed. First, the findings cannot be extended to urban areas in Saudi Arabia because the study was restricted to the rural Jazan area, but can be extended to other rural areas that are similar to the Jazan region in their criteria. Second, patients with HTN who did not visit the PHCCs were not covered in this study. Finally, the measurements of blood pressure were taken during morning visit. There was no option for home monitoring of blood pressure, and nocturnal variations and white-coat were not ascertained.

## 5. Conclusions and Recommendations

The adult patients who visited the selected PHCCs in rural Jazan had a reasonable awareness of some points regarding HTN risk factors but lacked the awareness of some other points, such as the normal BP and the fact that lifestyle changes could protect them from HTN or help in the treatment of the disease. Accordingly, healthcare providers in PHCCs should be trained to implement a health educational program to improve HTN awareness and knowledge.

## Figures and Tables

**Table 1 healthcare-11-01676-t001:** Background characteristics and prevalence and factors of hypertension among the adult population in rural Jazan.

Variables		All Participants (N = 607)		Hypertension		*p*-Value
			Normotensive (n = 379)	Hypertensive (n = 210)		
		n (%)	n (%)	n (%)		
Sex	Male	362 (59.6)		258 (71.3)	104 (28.7)	**<0.001**
	Female	245 (40.4)		139 (56.7)	106 (43.3)	
Age (year)	18–24	62 (10.2)		60 (96.8)	2 (3.2)	**<0.001**
	25–44	284 (46.8)		237 (83.5)	47 (16.5)	
	45–64	180 (29.7)		86 (47.8)	94 (52.2)	
	≥65	81 (13.3)		14 (17.3)	67 (82.7)	
Educational level	Illiterate	107 (17.6)		32 (29.9)	75 (70.1)	**<0.001**
	Primary	127 (20.9)		70 (55.1)	57 (44.9)	
	Secondary	173 (28.5)		142 (82.1)	31 (17.9)	
	University	200 (32.9)		153 (76.5)	47 (23.5)	
Income	SAR <5000	363 (59.8)		215 (59.2)	148 (40.8)	**0.002**
	SAR 5000–9999	125 (20.6)		93 (74.4)	32 (25.6)	
	SAR 10,000–14,999	70 (11.5)		53 (75.7)	17 (24.3)	
	SAR ≥15,000	49 (8.1)		36 (73.5)	13 (26.5)	
Marital status	Single	145 (23.9)	120 (82.8)	25 (17.2)	**<0.001**	
	Married	404 (66.6)	258 (63.9)	146 (36.1)		
	Divorced	23 (3.8)	13 (56.5)	10 (43.5)		
	Widowed	35 (5.8)	6 (17.1)	29 (82.9)		
Tobacco chewing	No	535 (88.1)	364 (68.0)	171 (32.0)	**<0.001**	
	Yes	63 (10.4)	28 (44.4)	35 (55.6)		
	Former user	9 (1.5)	5 (55.6)	4 (44.4)		
Smoking	Nonsmoker	485 (79.9)	312 (64.3)	173 (35.7)	0.284	
	Smoker	94 (15.5)	68 (72.3)	26 (27.7)		
	Ex-smoker	28 (4.6)	17 (60.7)	11 (39.3)		

The *p*-value was calculated using the chi-squared test; SAR = Saudi riyal. Bold values indicate statistical significance at the *p* < 0.05 level.

**Table 2 healthcare-11-01676-t002:** Awareness about hypertension and its risk factors among the rural Jazan population.

Statements	Normotensive (n = 379)			Hypertensive (n = 210)			*p*-Value
	Yes	No	I Do Not Know	Yes	No	I Do Not Know	
	n (%)	n (%)	n (%)	n (%)	n (%)	n (%)	
I am knowledgeable about the normal BP.	172 (43.3)	225 (56.6)	0 (0)	92 (43.8)	118 (56.1)	0 (0)	0.909
HTN can progress with age.	277 (69.7)	43 (10.8)	77 (19.4)	173 (82.3)	11 (5.2)	26 (12.3)	**0.003**
Medications are enough to control HTN.	64 (16.1)	245 (61.7)	88 (22.1)	44 (20.9)	124 (59.0)	42 (20.0)	0.322
HTN is a treatable disease.	241 (60.7)	69 (17.3)	87 (21.9)	136 (64.7)	27 (12.8)	47 (22.3)	0.341
Lifestyle changes have no role in controlling HTN.	87 (21.9)	211 (53.1)	99 (24.9)	61 (29.0)	94 (44.7)	55 (26.1)	0.087
HTN may lead to other life-threating disease.	318 (80.1)	14 (3.5)	65 (16.3)	177 (84.2)	3 (1.4)	30 (14.2)	0.242
Smoking is a risk factor for HTN.	318 (80.1)	30 (7.5)	49 (12.3)	158 (75.2)	17 (8.0)	35 (16.6)	0.313
Obesity is a risk factor for HTN.	345 (86.9)	29 (7.3)	23 (5.7)	186 (88.5)	10 (4.7)	14 (6.6)	0.450
Salt consumption is a risk factor for HTN.	347 (87.5)	24 (6.00)	26 (6.5)	191 (90.9)	8 (3.8)	11 (5.2)	0.390
Stress is a risk factor for HTN.	293 (73.8)	38 (9.5)	66 (16.6)	156 (74.2)	12 (5.7)	42 (20.0)	0.186
Red meat consumption is a risk factor for HTN.	191 (48.1)	72 (18.1)	134 (33.7)	110 (52.3)	28 (13.3)	72 (34.2)	0.295

The *p*-value was calculated using the chi-squared test; HTN = hypertension; BP = blood pleasure. Bold values indicate statistical significance at the *p* < 0.05 level.

**Table 3 healthcare-11-01676-t003:** Determinants of HTN.

Variables	I Am Knowledgeable about the Normal BP.	HTN Can Progress with Age.	Medical Treatment Is Enough to Control HTN.	HTN Is Treatable.	Lifestyle Changes Have No Role in Controlling HTN.	HTN May Lead to Other Life-Threating Disease.	Smoking Is a Risk Factor for HTN.	Obesity Is a Risk Factor for HTN.	Salt Consumption Is a Risk Factor for HTN.	Stress Is a Risk Factor for HTN.	Red Meat Consumption Is a Risk Factor for HTN.
Sex											
Male	137 (37.8)	266 (73.5)	60 (16.6)	244 (61.9)	91 (25.1)	298 (82.3)	290 (80.1)	321 (88.7)	313 (86.5)	252 (69.6)	178 (49.2)
Female	127 (51.8)	184 (75.1)	48 (19.6)	153 (62.4)	57 (23.3)	197 (80)	186 (75.9)	210 (85.7)	225 (91.8)	197 (80.4)	123 (40.2)
*p*-value	**<0.001**	0.855	0.424	0.130	0.732	0.290	0.339	0.214	0.086	**0.012**	0.226
Age (year)											
18–24	16 (25.8)	25 (40.3)	9 (14.5)	28 (45.2)	13 (21)	42 (67.7)	37 (59.7)	43 (69.4)	43 (69.4)	40 (64.5)	24 (38.7)
25–44	142 (50)	214 (75.4)	48 (16.9)	183 (64.4)	65 (22.9)	237 (83.5)	238 (83.8)	255 (89.8)	256 (90.1)	225 (79.2)	133 (46.8)
45–64	75 (41.7)	142 (78.9)	28 (15.6)	119 (66.1)	42 (23.3)	151 (83.9)	140 (77.8)	161 (89.4)	164 (91.1)	123 (68.3)	104 (57.8)
≥65	31 (38.3)	69 (85.2)	23 (28.4)	47 (58)	28 (34.6)	65 (80.2)	61 (75.3)	72 (88.9)	75 (92.6)	61 (75.3)	40 (49.4)
*p*-value	**0.003**	**<0.001**	**0.020**	0.056	0.324	0.149	**<0.001**	**<0.001**	**<0.001**	**0.002**	**0.022**
Educational level											
Illiterate	22 (20.6)	85 (79.4)	29 (27.1)	66 (61.7)	32 (29.9)	78 (72.9)	76 (71.0)	94 (87.9)	99 (92.5)	75 (70.1)	49 (45.8)
Primary	46 (36.2)	94 (74.0)	22 (17.3)	79 (62.2)	30 (23.6)	97 (76.3)	92 (72.4)	107 (84.2)	104 (81.8)	91 (71.6)	64 (50.3)
Secondary	74 (42.8)	118 (68.2)	25 (14.5)	112 (64.7)	45 (26.0)	146 (84.4)	138 (79.8)	148 (85.5)	150 (86.7)	111 (64.2)	85 (49.1)
University	122 (61.0)	153 (76.5)	32 (16.0)	120 (60.0)	41 (20.5)	174 (87.0)	170 (85.0)	182 (91.0)	185 (92.5)	172 (86.0)	103 (51.5)
*p*-value	**<0.001**	0.131	**<0.001**	0.524	**0.002**	0.032	**0.034**	**0.018**	0.051	**<0.001**	0.680
Income (SAR)											
<5000	116 (32.0)	263 (72.5)	63 (17.4)	219 (60.3)	90 (24.8)	287 (79.1)	267 (73.6)	314 (86.5)	316 (87.1)	257 (70.8)	173 (47.7)
5000–9999	66 (52.8)	89 (71.2)	22 (17.6)	83 (66.4)	30 (24.0)	98 (78.4)	104 (83.2)	106 (84.8)	110 (88.0)	85 (68.0)	57 (45.6)
10,000–14,999	46 (65.7)	59 (84.3)	14 (20.0)	48 (68.6)	19 (27.1)	65 (92.9)	62 (88.6)	66 (94.3)	64 (91.4)	61 (87.1)	45 (64.3)
≥15,000	36 (73.5)	39 (79.6)	9 (18.4)	27 (55.1)	9 (18.4)	45 (91.8)	43 (87.8)	45 (91.8)	48 (98.0)	46 (93.9)	26 (53.1)
*p*-value	**<0.001**	0.215	0.332	**0.047**	**<0.001**	0.060	**0.011**	0.370	0.118	**<0.001**	0.097
Marital status											
Single	53 (36.6)	89 (61.4)	34 (23.4)	85 (58.6)	34 (23.4)	111 (76.6)	108 (74.5)	115 (79.3)	118 (81.4)	105 (72.4)	65 (44.8)
Married	186 (46)	314 (77.7)	59 (14.6)	255 (63.1)	86 (21.3)	332 (82.2)	323 (80)	366 (90.6)	367 (90.8)	297 (73.5)	211 (52.2)
Divorced	10 (43.5)	17 (73.9)	8 (34.8)	17 (73.9)	11 (47.8)	22 (95.7)	18 (78.3)	19 (82.6)	21 (91.3)	18 (78.3)	6 (26.1)
Widowed	15 (42.9)	30 (85.7)	7 (20)	20 (57.1)	17 (48.6)	30 (85.7)	27 (77.1)	31 (88.6)	32 (91.4)	29 (82.9)	19 (54.3)
*p*-value	0.271	**0.004**	**0.027**	**0.045**	**<0.001**	0.137	0.435	**0.010**	**0.047**	0.351	**0.029**

The *p*-value was calculated using the chi-squared test; SAR = Saudi riyal; HTN = hypertension. Values are presented as frequencies and percentages. Bold values indicate statistical significance at the *p* < 0.05 level.

**Table 4 healthcare-11-01676-t004:** Logistic regression analysis of the independent predictors of hypertension.

Characteristics		Univariate				Multivariate			
		cOR	95% CI		*p*-Value	aOR	95% CI		*p*-Value
			Lower	Upper			Lower	Upper	
**Age (Year)**		**1.12**	**1.10**	**1.15**	**<0.001**	1.13	1.10	1.16	**<0.001**
Sex	Male *	1				1			
	Female	1.83	1.29	2.59	**0.001**	1.79	1.06	3.03	**0.029**
Educational level	Illiterate *	1							
	Primary	0.28	0.15	0.52	**<0.001**	1.11	0.48	2.57	0.816
	Secondary school	0.07	0.03	0.12	**<0.001**	1.46	0.56	3.80	0.440
	University	0.09	0.05	0.15	**<0.001**	1.85	0.69	4.96	0.225
Income	SAR <5000 *	1				1			
	SAR 5000–9999	0.466	0.294	0.739	**0.001**	0.49	0.26	0.93	**0.029**
	SAR 10,000–14,999	0.424	0.235	0.768	**0.005**	0.50	0.21	1.16	0.107
	SAR ≥15,000	0.454	0.232	0.89	**0.021**	0.36	0.14	0.92	**0.032**
Marital status	Single *	1				1			
	Married	2.716	1.687	4.374	**<0.001**	0.98	0.51	1.90	0.951
	Divorced	3.692	1.456	9.360	**0.006**	1.91	0.61	6.03	0.269
	Widowed	23.200	8.716	61.752	**<0.001**	1.49	0.34	6.63	0.598
Tobacco chewing	No *	1				1			
	Yes	4.22	2.27	7.85	**<0.001**	1.62	0.66	3.96	0.290
	Former user	2.57	0.57	11.63	0.219	4.40	0.41	47.29	0.222

* Reference category; CI = confidence interval; cOR = crude odds ratio; aOR = adjusted odds ratio; SAR = Saudi riyal. Bold values indicate statistical significance at the *p* < 0.05 level.

## Data Availability

Data used in this study is available.

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
