# Peer review of "Prevalence and Awareness of Hypertension among a Rural Jazan Population"

_healthcare, 2023, doi:10.3390/healthcare11121676_

Round 1
Reviewer 1 Report
In this study by Luai Alhazmi and colleagues, the authors detected the prevalence and awareness about hypertension among rural jazan population. The big problem of this study is some conclusions are not supported by sufficient evidence. I have a few concerns about the following issues which may help to strengthen the conclusions and improve the scientific rigor.
1. Importantly, please double-check the statistical analysis methods and the descriptions. a. Please clear what tests are you using; b, for continuous variables, please make sure normality and equal variance were checked before analysis.
2. Table3: Please modify the table to improve the readability, for example move the variables to the left..
3. The conclusion about that “women 11 have a higher prevalence of HTN (43.3%) than men (34.6%) in the rural Jazan region”: I don't think it‘s rigorous since the study only include those patients who visited these centers, and you mentioned in the story that the awareness of hypertension in rural areas is not high. Please re-consider your conclusion.
4. The conclusion about “the prevalence of HTN was inversely related to wealth”, “a statistically significant correlation between participant level of education and the level of knowledge about HTN” also need further verification by using correlation analysis.
5. The conclusion about “Our survey found that rural Jazan adults were more aware about hypertension risk factors than those in other studies conducted around the world.” also need further study to verify since you only include those people who visited the centers.
6. In the limitation, “so this study is covering the low and middle social classes and few or none of high social class Saudis, however most of those living in the rural areas are low and middle social classes.” How you made this conclusion? Please clarify.
7. As a clinic study, ethics approval and consent to participate should be mentioned in the method.
1. Proof-reading for the language is needed, including the grammatical errors and some spelling errors. Also, terms should be defined the first time they are mentioned.
Author Response
Reply attached

Reviewer 2 Report
The article presented by Alhazmi et al. describes a study aimed at understanding and assessing hypertension among the rural population of rural population of Jazan, Saudi Arabia. The study used interview questionnaires completed by 607 people attending six primary health care centres in the region.
In general, the following suggestions could help strengthen the argument, relevance, and potential impact of the article.
Section 1. Introduction
Overall, the introduction provides a good overview of hypertension and its impact on public health, as well as the need to raise awareness and knowledge about the disease, especially in rural areas such as Jazan, Saudi Arabia. However, the introduction should be expanded to include more literature references and there are some aspects that could be improved:
· This section should be better organised to clearly present the different aspects of the problem, such as the causes and risk factors of hypertension, the global and local prevalence of the disease and the need for increased awareness.
· More specific information should be provided on the determinants of hypertension awareness among the rural population of Jazan, Saudi Arabia, and how this study will contribute to addressing this problem.
· More explicit statements of the main objectives and research questions of the study
Section . 2. Methodology
Authors should include a brief explanation as it would be useful to know if there were any inclusion or exclusion criteria other than those mentioned in section 2.2. For example, were there any specific diseases or medications that might have influenced the results and were therefore excluded from the study?
The authors indicate below that validated questionnaires were used for data collection. They should indicate this. They should also indicate the equipment used for blood pressure measurement.
It could be helpful to include information about the process of data analysis. For example, the statistical methods used to analyze the collected data, such as descriptive statistics, chi-square test, or logistic regression. Additionally, it could be helpful to include information on how missing data or incomplete responses were handled in the analysis.
Finally, authors should note that it is vitally important to include a statement on ethical considerations, such as obtaining informed consent from participants and approval from an institutional review board (IRB) or ethics committee. Indicate the IRB code.
Section . 3. Results
Authors should include the comparison of the populations within each group, not just the overall p-value. They should provide a p for each stratum.
The type of statistical test used should be indicated at the bottom of the table.
Acronyms should be defined below each table.
Bold the values that are statistically significant to make the table easier to read.
In section 3.3 they should include the n within the parentheses. It is confusing.
In table 3 the variables should be at the beginning to make the table easier to read. Homogenize the data. At the foot of the table indicate the data being expressed.Section 4. Discussion
Overall, the discussion could be improved by focusing on the implications of the findings, addressing potential sources of bias, and providing more up-to-date references.
Some of the references cited in the discussion are quite old, which may suggest that more recent research in this area has not been considered. The discussion could benefit from more up-to-date references to strengthen its argument and relevance.
Section 5. Conclusion and recommendations.
The authors should provide more specific and practical recommendations to health professionals. For example, what should the health education programme include, how often, should it be more targeted to certain age groups or gender?
The authors should address potential barriers or difficulties that may arise in implementing the proposed interventions, such as limited resources or patient resistance.
Identify areas for future research, such as exploring the effectiveness of different types of health education programmes or examining the impact of increased awareness and knowledge on hypertension-related outcomes over time.
Author Response
Replay attached. Please notice that due to major revision, as advised, the updated manuscript was uploaded to the dashboard with all necessary editing.

Round 2
Reviewer 1 Report
No further comments.
Reviewer 2 Report
Authors should attach the response to the proposed comments.
